# Key Signaling Pathways in Aging and Potential Interventions for Healthy Aging

**DOI:** 10.3390/cells10030660

**Published:** 2021-03-16

**Authors:** Mengdi Yu, Hongxia Zhang, Brian Wang, Yinuo Zhang, Xiaoying Zheng, Bei Shao, Qichuan Zhuge, Kunlin Jin

**Affiliations:** 1Zhejiang Provincial Key Laboratory of Aging and Neurological Disorder Research, The First Affiliated Hospital of Wenzhou Medical University, Wenzhou 325000, China; 15726898102@163.com (M.Y.); yinuo_zhang818@163.com (Y.Z.); evezxy97@163.com (X.Z.); 2Department of Pathology, University of Texas Southwestern Medical Center, Dallas, TX 75390, USA; Hongxia.Zhang@utsouthwestern.edu; 3Pathnova Laboratories Pte. Ltd. 1 Research Link, Singapore 117604, Singapore; brian@pathnova.com; 4Department of Neurology, The First Affiliated Hospital of Wenzhou Medical University, Wenzhou 325000, China; shaobei56@126.com; 5Department of Pharmacology and Neuroscience, University of North Texas Health Science Center, Fort Worth, TX 76107, USA

**Keywords:** AMPK, SIRT1, mTOR, aging, senescence, health span, intervention, signaling

## Abstract

Aging is a fundamental biological process accompanied by a general decline in tissue function. Indeed, as the lifespan increases, age-related dysfunction, such as cognitive impairment or dementia, will become a growing public health issue. Aging is also a great risk factor for many age-related diseases. Nowadays, people want not only to live longer but also healthier. Therefore, there is a critical need in understanding the underlying cellular and molecular mechanisms regulating aging that will allow us to modify the aging process for healthy aging and alleviate age-related disease. Here, we reviewed the recent breakthroughs in the mechanistic understanding of biological aging, focusing on the adenosine monophosphate-activated kinase (AMPK), Sirtuin 1 (SIRT1) and mammalian target of rapamycin (mTOR) pathways, which are currently considered critical for aging. We also discussed how these proteins and pathways may potentially interact with each other to regulate aging. We further described how the knowledge of these pathways may lead to new interventions for antiaging and against age-related disease.

## 1. Introduction

Nowadays, the population and life expectancy of humans are significantly increasing, due in large part to improvements in nutrients, medicine and environments, etc., which results in a decrease in mortality from age-related diseases, such as heart disease, cancer and stroke [1]. The world’s total population was about 6.76 billion in 2008, and the number increased to 7.59 billion in 2018. Sixteen percent will be over the age of 65 by 2050, up from 9% in 2019 in the world, suggesting that the age group of 65 and over is growing the fastest. Aging is a fundamental biological process accompanied by a general decline in tissue function and increased risk for many age-related diseases. According to the National Council on Aging, about 92 percent of the elderly have at least one age-related disease and 77% have at least two. Heart disease, ischemic stroke, cancer and diabetes are among the most common disorders. For example, as the lifespan increases, cognitive impairment or dementia will become a growing public health issue. According to current estimates, almost 36 million people have dementia worldwide, and this number is expected to reach 66 million by 2030 and 115 million by 2050 [2]. The population with dementia is expected to increase to a total of over 13 million in the United States alone. The most well-known form of dementia is Alzheimer’s disease (AD), but a large percentage of aged cognitively impaired persons are not due to AD but, rather, the normal aging process. The vast majority of older adults suffer declines in cognitive functions, interfering with their ability to participate and engage in meaningful activities [3]. In addition to AD, deterioration in fine motor control, gait and balance are among the most important health problems in the elderly. Now, falls are the leading cause of injury-related death and the third-leading cause of poor health among persons aged 65 years and older [4]. In 2013, the cost of these injuries caused by falls was US $34 billion. Thus, there is a critical need in understanding underlying cellular and molecular mechanisms regulating aging, which will allow us to modify the aging process for healthy aging and alleviate age-related disease.

This review focused on the molecular mechanisms involved in biological aging—specifically, the adenosine monophosphate-activated kinase (AMPK), Sirtuin 1 (SIRT1) and mammalian target of rapamycin (mTOR) pathways; these are currently considered the critical signaling pathways for the aging process. We also discussed how the knowledge of these pathways may lead to new interventions for antiaging and against age-related disease.

## 2. AMPK Signaling

AMPK (a serine/threonine protein kinase) is an essential energy sensor engaged in modulating our whole-body level of metabolic energy balance [5,6]. In mammals, AMPK comprises of two isoforms of the α and β subunits each and three isoforms of the γ subunits to form a combination of 12 different αβγ isoforms. The α subunit encodes an N-terminal protein kinase domain, which interacts with a C-terminal regulatory domain via its well-known activation segment threonine 172 (T172) to mediate AMPK’s catalytic activity. The β subunit has a carbohydrate-binding module (CBM), and its C-terminal domain connects the γ subunit and the C-terminal domain of the α subunit. The γ subunit holds four cystathionine-β-synthase (CBS) regions referring to adenine nucleotide binding, which ensures the monitoring and regulation for AMP/ATP ratio [7,8,9]. Both the β (β1 and β2) and γ (γ1, γ2 and γ3) subunits regulate AMPK’s phosphorylation and activity. Traditionally, AMPK signaling is one of the central regulators of cellular and organismal metabolism in eukaryotes, playing important roles in regulating growth and reprogramming metabolism. Recently, AMPK signaling has been connected to aging and the lifespan, as AMPK can control the regulation of cellular homeostasis, resistance to stress, cell survival and growth, cell death and autophagy. Supportively, specific AMPK activation protects against aging and extends the lifespan in *Caenorhabditis elegans* (*C. elegans*) and rodents.

### 2.1. AMPK Signaling and Aging

AMPK signaling is predominantly activated by three main upstream kinases, including liver kinase B1 (LKB1), also known as serine/threonine kinase 11 (STK11), Ca^2+^/calmodulin-dependent protein kinase kinase β (CaMKKβ) and transforming growth factor β-activated kinase 1 (TAK1). Other upstream molecules such as protein phosphatase 2Cα (PP2Cα) have relatively weaker function on activating AMPK [8,9]. Both LKB1 and CaMKKβ can activate AMPK by phosphorylating T172. LKB1’s expression increases in tandem with an increase in the AMP/ATP ratio, while CaMKKβ’s activity increases via the production of reactive oxygen species (ROS) and increased intracellular Ca^2+^ level induced by inflammatory stimuli [7,8,9]. Downstream mediators of AMPK signaling include mTOR, aminocyclopropane-1-carboxylic acid (ACC1), glucose transporter 1 (GLUT1)/GLUT4, p53, autophagy activating kinase 1/2 (ULK1/2), peroxisome proliferator-activated receptor gamma coactivator-1α (PGC1-α) and forkhead box transcription factors (FOXOs), which are components of the central metabolic activities in the aging process [8] (Figure 1). On the cellular level, the LKB1/AMPK signaling pathway plays a critical role in cell proliferation, migration and apoptosis by inhibiting myosin light chain kinase (MLCK) and the phosphorylated myosin light chain (p-MLC) induced by quercetin, a common flavonoid in many fruits and red wine [10]. In 3T3-L1 adipocytes, testosterone (100 mM) increased GLUT4-dependent glucose uptake through the LKB1/AMPK signaling pathway [11]. Estradiol (E2) is an important sex hormone protecting females from aging, especially in cardiovascular diseases. In C2C12 myotubes, E2 can activate AMPK by interacting with estrogen receptors (ERs) through its metabolite 2-hydroxyestradiol (2-HE2) [12]. Another study found that E2 and its mimetic nordihydroguaiaretic (NDGA) inhibited angiotensin II-induced vascular smooth muscle cell (VSMC) proliferation by increasing the expression of ERα, AMPK and LKB1 [13]. On the organismal level, the double-knockdown of protein kinase B 2 (Akt2) and AMPK contributed to heart aging without influencing the lifespan, suggesting that AMPK may protect against impaired autophagy and mitophagy in aging [14]. Strawberries significantly improved the antioxidant activity and mitochondrial biomass and function in aged rats through the AMPK pathway [15]. In mice lacking the growth hormone receptor gene, males had higher expression levels of AMPK, PGC1-α and SIRT1 than females in the brain and kidneys, indicating their key roles in mitochondrial biogenesis [16]. A methionine-restricted diet could restore young metabolic phenotype in adult mice by improving serum fibroblast growth factor 21 (FGF-21) levels to increase the expression of mitochondrial enzymes and further increase AMP/ATP level, which, in turn, activate AMPK [17]. β-adrenergics could accelerate myocardial fibrosis by downregulating AMPK activity and upregulating β-arrestin 1 in aged mice [18]. AMPK signaling also participates in the age-related decline of hippocampal neurogenesis [6], which has been shown to contribute to cognitive impairment. Taken together, these findings suggest that age-related alterations are associated with the AMPK pathway. However, whether AMPK activity is altered with age remains a hot topic of debate. Several studies suggest that AMPK activity is increased, whereas other data suggest that it is decreased with aging [6,19,20].

### 2.2. Metformin and AMPK Signaling

Metformin, a known AMPK activator, is a United States Food and Drug Administration (FDA)-approved first-line oral blood glucose-lowering prescribed drug for the treatment of type 2 diabetes, which was first isolated from the French lilac, an ancient herbal remedy in the 17th century [21,22]. In the early 2000s, reports from the National Institutes of Health (NIH) found that metformin could extend the lifespan and health span in mice and decrease the risk of cancer and other age-related diseases. Consistently, metformin also extended the health span and median lifespan, as well as improved the youthful mobility in *C. elegans*, which was first reported in 2010 [23,24]. In humans, the first randomized controlled clinical study named the Targeting Aging with Metformin (TAME) evaluated metformin’s antiaging capacity, such as delaying the occurrence and progression of age-related diseases [25,26]. Currently, there are 16 clinical trials focused on metformin and aging and longevity (http://clinicaltrials.gov/ct2/home). To date, metformin is regarded as a caloric restriction mimetic (CRM), which is defined as a series of nontoxic compounds that can reduce the level of protein acetylation and induce autophagy [27]. Metformin can directly or indirectly activate AMPK by decreasing the ADP/ATP and AMP/ATP levels through the partial inhibition of Complex I of the mitochondrial election transport chain (ETC) [28,29]. In Hutchinson-Gilford progeria syndrome (HGPS), metformin can ameliorate accelerated aging defects through cellular stress-induced AMPK activation [30]. In aged skin, the local application of metformin promoted wound healing and cutaneous integrity by improving vascularization through the AMPK pathway [31]. Metformin also activates AMPK signaling to reduce age-related hepatic lipid accumulation, remodel extracellular matrix in adipose tissue and decrease insulin resistance in obesity by inhibiting the downstream transforming growth factor-β1 (TGFβ1)/Smad3 signaling and the upstream Rho-kinase 1 (ROCK1), which has been shown to accelerate obesity-induced steatosis in mice [32,33]. Further, metformin improves the age-related changes in liver sinusoidal endothelial cells via the AMPK and endothelial nitric oxide (NO) pathways to increase ATP, cyclic guanosine monophosphate (cGMP) and mitochondrial activity [34]. Metformin also attenuates age-related hearing loss, cell apoptosis and neurodegeneration in d-galactose-induced aging rats through the unfolded protein response (UPR) via the AMPK/extracellular signal-regulated kinase 1/2 (ERK1/2) signaling pathways [35]. Notably, clinically relevant concentrations of metformin were able to protect PC12 cells and hippocampal neurons from oxidative injury via the activation of AMPK [36]. In Parkinson’s disease (PD) and normal aging, metformin controlled the astrocyte activity by inducing the activation of AMPK and brain-derived neurotrophic factor (BDNF) signaling [37]. Metformin also inhibits the phosphorylation and accumulation of α-synuclein, to protect from mitochondrial dysfunction and oxidative stress, regulate autophagy and attenuate neurodegeneration and neuroinflammation via the activation of AMPK signaling [38] (Table 1). Although metformin has many benefits on aging via AMPK signaling, its application for antiaging remains restricted. For instance, there are no evidence supporting these effects on normal animals and humans up until now, and its lactic acidosis side effect is worth paying attention to.

### 2.3. Resveratrol and AMPK Signaling

Resveratrol (also named as 3, 5, 4′-trihydroxy-transstilbene) is a polyphenol found in some plants like berries, nuts and grapes [52]. It was first reported to increase the yeast lifespan in 2003 [53]. Resveratrol can activate AMPK when the intracellular calcium level increases [54]. AMPK activation can prevent the production of endogenous ROS to improve osteogenic differentiation in aged bone mesenchymal stem cells [39,55]. Huang et al. reported that resveratrol can reverse mitochondrial dysfunction and oxidative stress through the protein kinase A (PKA)/LKB1/AMPK pathway in high-fat diet-induced muscle atrophy in aged rats [39]. Moreover, resveratrol was reported to decrease the expression of intercellular adhesion molecule-1 (ICAM-1) and monocyte adhesiveness to tumor necrosis factor α (TNFα)-treated endothelial cells through an anti-inflammatory cascade triggered by miR-221/222/AMPK/p38/NF-kappa B (NF-κB) pathway [56]. In cultured primary human keratinocytes, resveratrol activated AMPK/FOXO3 signaling to guard against oxidative stress-induced senescence and proliferative impairment [57]. In addition, resveratrol could activate AMPK via LKB1 to stimulate mitochondrial biogenesis in neurons, resulting in neuroprotective effects and the inhibition of protein synthesis and genetic transcription by inactivating Akt and, thus, prevent cardiac hypertrophy [58,59]. However, the antiaging mechanisms of resveratrol are not only mediated by AMPK but also SIRT1 and mTOR, which will be discussed later.

### 2.4. Physical Exercise, a Link between AMPK Signaling and Aging

For many years, it has been well-known that regular physical exercise can attenuate the major hallmarks of aging and promote health span by slowing down age-related degenerative processes [60,61,62]. Indeed, as we age, the loss of muscle mass and the risks of heart disease and dementia increases, and immune function becomes increasingly impaired [63]. For example, human aging is associated with skeletal muscle atrophy and functional impairment (sarcopenia), which can be partially reversed following six months of resistance exercise training [64]. Interestingly, a recent study showed that consistent aerobic exercise may not only slow down the effects of aging but results in rejuvenation; aged mice with voluntary exercise improved muscle stem cell function, which is inefficient in aged organisms and accelerated muscles tissue repair [65]. Although the precise mechanism remains largely unexplored, AMPK signaling plays an important role in exercise-mediated reversal aging effects.

Liu et al. reported that regular aerobic physical activity induced autophagy in the hippocampus of middle-aged rats by regulating calmodulin-dependent protein kinase (CAMK)/AMP/AMPK/Beclin 1 pathways [66]. Following a ten-week regular moderate intensity aerobic exercise, the AMPK activation was increased in parallel with the decreased age-related autophagy and cell apoptosis [67]. Long-term high-intensity interval training (HIIT), on the other hand, upregulates adiponectin/AMPK signaling to improve autophagy, oxidative stress, mitochondrial function and apoptosis in the skeletal muscles of aged female rats [68]. Yoon et al. found that AMPK activation induced by exercise protected the skeletal muscle from senescence including muscle size, mass and strength [69]. In d-galactose-induced aged rats, exercise coupled with a spermidine supplement mitigated skeletal muscle atrophy by enhancing autophagy and reducing apoptosis through the AMPK/FOXO3 pathway [70]. Dehydroepiandrosterone (DHEA) would increase with caloric restriction or exercise and can further increase the activities of AMPK, PGC1-α and GLUT4 [71]. In a clinical study involving ten young volunteers, AMPK activation induced by cycling endurance exercise increased telomeric repeat-containing RNA (TERRA), suggesting that exercise might counteract the decline of telomere integrity resulting in antiaging [72].

### 2.5. Autophagy and AMPK Signaling in Aging

Autophagy plays a critical role in the resistance of senescence by removing metabolic products as well as injured cells or organelles [73]. AMPK protects cells from oxidative stress-induced senescence by restoring the autophagic flux and NAD+ level in aged cells through the NAD+ synthetic rescue method [74]. The AMPK signaling activation ameliorated autophagy and the level of coactivator-associated arginine methyltransferase 1 (CARM1), which involves several cellular processes, including autophagy in the aging heart [75]. Metformin can stimulate autophagy via activating AMPK signaling to protect human retinal pigment epithelial cells from hydrogen peroxide induced-oxidative stress [76]. In a d-galactose-induced renal aging model, hyperoxide attenuated a renal injury by augmenting autophagy through AMPK/ULK1 signaling [77,78]. Oxidative stress-induced senescence could result in the abnormal aging of skin cells, and caffeine protected against ultraviolet irradiation-induced senescence in mouse skin tissues by activating autophagy, which accelerated the elimination of ROS through the adenosine A2a receptor (A2AR)/SIRT3/AMPK pathways [79]. Kim et al. reported that poria cocus wolf extract activated autophagy, regulated lipid metabolism and inhibited endoplasmic reticulum (ER) stress by AMPK activation and thereby prevented liver hepatic steatosis [80]. Natural triterpene saponins complex aescin increased the intracellular ROS levels and activated the ataxia–telangiectasia-mutated kinase (ATM)/AMPK/ULK1 pathway to regulate autophagy [81]. A creatine analog, β-guanidinopropionic acid (β-GPA), prolonged the lifespan of Drosophila melanogaster by increasing autophagy via the elevation of AMPK activation levels [82]. AMPK/p27 signaling balanced the autophagy and apoptosis, the two main components of programmed cell death, by inhibiting the decline of aged skeletal muscle stem cell function [83,84]. The NF-E2-related factor 2 (NRF2) transcription factor, one of the central regulators of oxidative stress and electrophilic stress responses, suppressed autophagy by reducing the expression of AMPK during chronic oxidative stress [85]. In all, these data demonstrate the effects of AMPK signaling on autophagy in antiaging.

## 3. Sirtuin Signaling

Sirtuin (SIRT) was first discovered in the 1970s in nature and is an essential factor in delaying cellular senescence and extending organismal lifespan [86]. SIRT1 is the most well-studied among the seven SIRT isoforms in humans [87]. SIRT1 is an NAD+-dependent deacetylase and, thus, can deacetylate tumor suppressor p53 protein [88,89], the DNA repair factor Ku70 [90], NF-κB [91], the signal transducer and activator of transcription 3 (STAT3) [92] and the FOXO family of forkhead transcription factors (Figure 2). The roles of Sirtuin on the suppression of cellular senescence is primarily mediated through delaying age-related telomere attrition, sustaining genome integrity and promoting DNA damage repair. SIRT1 can improve the ability to induce cell cycle arrest and oxidative stress resistance and inhibits cell death [93] and apoptotic pathways [94]. It has been verified that Ku70 blocks the death of stress-induced apoptotic cells by sequestering the proapoptotic factor Bax away from the mitochondria. NF-κB is involved in upregulating gene products controlling cell survival. SIRT1 participates in many age-related processes and disorders, such as neurodegenerative diseases and cardiovascular diseases, etc. [95].

### 3.1. SIRT1 Signaling and Aging

Sirtuins are reported to exert a prolongevity effect of budding yeast S. cerevisiae, nematode *C. elegans*, fruit flies *D. melanogaster* and mice when the expression levels of Sirtuin, especially SIRT2 and SIRT6, are increased [96,97,98,99]. Similarly, brain-specific Sirt1-overexpressing (BRASTO) transgenic mice increased their median lifespan by 16% in females and 9% in males [100]. Sirtuin extends the organismal lifespan through the regulation of diverse cellular processes. SIRT1 activation can increase the sensitivity of insulin and reduce insulin resistance [87]. SIRT1 activators like resveratrol, SRT1720 and MHY2233 can improve insulin resistance and have beneficial effects on diabetes- or obesity-induced fatty liver [101]. Alkylresorcinols, a member belonging to the family of phenolic lipids, activated SIRT1-dependent deacetylation to reduce acetylated histone in human monocyte cells and prolong the lifespan of *D. melanogaster* [102]. Ursolic acid can directly stimulate SIRT1 by binding to the outer surface of SIRT1, further changing its structure from the inactive form to its active form, be it in silico, in vitro or in vivo, and plays a significant role in the aging process [103]. Dehydroabietic acid, a natural diterpene resin acid of confers, can directly activate SIRT1 to prevent lipofuscin accumulation and collagen secretion in humans and extend the lifespan in *C. elegans*, as reported by Kim et al. [104]. Pyridoxamine, an advanced glycation end product (AGE) inhibitor, could inhibit the accumulation of AGE and upregulate the expression of SIRT1 and ERα, as well as decrease the level of TGFβ in mesangial cells of 19-month-old ovariectomized female mice [105]. A03, an ApoE4-targeted SIRT1 enhancer, can elevate the expression level of SIRT1 in the hippocampus in 5xFAD-ApoE4 (E4FAD) AD mice to improve cognitive function [40]. 17β-estradiol activated ERα/SIRT1 to reduce oxidative stress, neuroinflammation and neuronal apoptosis in d-galactose-induced male mice and increased the SIRT1 level by enhancing the degradation of PPARγ via E3 ubiquitin ligase NEDD4-1 to delay cellular aging [106,107]. Although SIRT1 has many beneficial effects on aging and widely exists in our body, their mechanisms underlying antiaging remain unclear.

### 3.2. NAD+ and SIRT1 Signaling

Nicotinamide adenine dinucleotide (NAD+) was first discovered by Harden and Young in 1906 as a “cozymase” factor in fermentation, which could improve the rate of fermentation in yeast extracts [108]. Studies have demonstrated that high NAD+ levels could improve mitochondrial function, modulate DNA repair, reduce metabolic stress symptoms and improve other biological processes [109,110]. In 2012, Canto et al. first showed that nicotinamide riboside (NR), a NAD+ precursor, supplementation in mammalian cells and mice tissues increased the NAD+ level and activated SIRT1 and SIRT3 and eventually improved the oxidative metabolism and protected from high-fat diet-induced metabolic abnormalities [111]. In addition, NR could prevent and reverse nonalcoholic fatty liver disease (NAFLD) by inducing a SIRT1- and SIRT3-dependent mitochondrial unfolded protein response [112]. The NAD+ precursor nicotinamide mononucleotide (NMN) can suppress acute renal injury in a SIRT1-dependent manner and inhibit heart failure and DNA damage induced by radiation [108,113,114]. NMN also improved NO-mediated endothelium-dependent dilation (EDD) and reduced arterial oxidative stress by stimulating SIRT1 in the arteries [115]. Similarly, NMN can promote osteogenesis and decrease adipogenesis in aged mice and keep the telomere length from shortening to protect against telomere-dependent disorders via SIRT1 activation [116,117,118]. A transcriptome analysis suggested that NMN administration can reverse most of SIRT1-regulated genes expression induced by aging, e.g., through the increase in NAD+ levels to enhance the expression level of SIRT1 in the neurovascular unit by rejuvenating the mitochondria [119]. Mendelsohn et al. found that NMN or NR supplementation increased the lifespan in aged mice via the NAD+/poly-ADP-ribose polymerase 1 (PARP1)/SIRT1 axis [120]. Moreover, α7 nicotinic acetylcholine receptor can stimulate the NAD+/SIRT1 pathway to improve angiotensin II-induced senescence in VSMCs [121]. Certainly, SIRT1 can extend the lifespan by increasing the level of nicotinamide phosphoribosyl transferase (NAMPT), which is a critical enzyme needed for the production of NAD+ [122]. As we age, the levels of NAD+ will dramatically reduce, leading to the loss of Sirtuin and PARP activities. Although the mechanisms are unclear, it is not far-fetched to hypothesize that the synthesis of NAD+ does decline with age. However, a very recent study suggests that there is an increased degradation of NAD+ with age [123]. CD38 is known as a membrane-bound NADase that hydrolyzes NAD+ to nicotinamide and (cyclic-)ADP-ribose, and mice lacking CD38 or treated with the CD38 inhibitor experience an increase in the level of NAD+ [124]. Interestingly, the CD38 protein level increases in multiple tissues during aging, with a corresponding increase in CD38 enzymatic activity and declining NAD+ levels [123]. Importantly, CD38 not only degrades NAD+ but, also, NMN. In consideration of the origin of NAD+, the favored approach in humans is to supplement with NAD+ precursors to increase NAD+ and, thus, fight aging. These findings suggest that the efficacy of NAD+ precursors may be enhanced by co-supplementations with CD38 inhibitors such as thiazoloquin(az)olinones [125,126].

### 3.3. Resveratrol and SIRT1 Signaling

Early studies documented that resveratrol can extend the lifespan of budding yeast [53] and the honey bee [127]. However, further studies have shown that rats or mice treated with red wine or equivalent pharmacological doses of resveratrol do not extend the lifespan but do display an improvement of aging phenotypes [128,129]. Therefore, there is a hypothesis that resveratrol affects lifespan through SIRT signaling, as resveratrol is the first and representative Sirtuin activator [87], which depends on the dietary nutrient composition [130]. For instance, resveratrol activates SIRT1 to adjust the deacetylation status of core autophagy protein ATG9A so that the death of hair cells is reduced and age-related hearing loss is delayed [131]. The long-term treatment of resveratrol improves the exercise ability and voluntary motor behavior and reduces the negative changes in insulin and apoptotic signaling through the SIRT1/FOXO1 pathway in senescence-accelerated mouse prone 8 (SAMP8) mice [132]. Furthermore, resveratrol can influence bone homeostasis through SIRT1/eNOS/bone morphogenic protein 2 (BMP2) signaling [133]. The long-term supplementation of resveratrol can also delay senescence by the regulating senescence-associated secretory phenotype (SASP) through the SIRT1/nuclear-factor kappa B (NF-κB) signaling pathway [134]. Increased SIRT1 mRNA expression and decreased NF-κB expression were also found to modulate the aging-related status, inflammation, oxidative stress and apoptosis in the pancreas of old SAMP8 mice [135]. The short-term injection of resveratrol attenuated the oocyte degeneration in middle-aged mice by significantly ameliorating oxidative stress through the increased expression of SIRT1 to reduce ROS and promote mitochondrial function [136]. Ham et al. found that resveratrol reduced ROS generation in the nematode via the SIRT1/FOXO3a axis [137]. By enhancing SIRT1 level and antioxidant production, resveratrol protected myoblasts from high ROS-induced apoptosis [41]. By stimulating the Akt/eNOS/NO and SIRT1/ER pathways, resveratrol was protected from diabetes-caused vascular disorders in C57BL/6 male mice fed with a 17-week high-fat diet [138]. In Machado-Joseph disease (MJD), a neurodegenerative disease with an abnormal expansion of the CAG triplet in the ATXN3 gene, resveratrol elevated the SIRT1 expression to improve motor deficits [46]. Resveratrol also improved learning and spatial memory in Htau mice (a model of AD) by activating SIRT1, leading to increase tau exon 10 inclusion and further expressing 4R-tau [41]. Enhancing p-CREB and SIRT1 protein by resveratrol markedly increased the number of newly generated cells in the hippocampus in aged rats [139]. Resveratrol-loaded nano-emulsion can also protect aged rats with abdominal surgery from cognitive dysfunction by SIRT1 signaling [140].

### 3.4. Exercise and SIRT1 Signaling

Exercise has positive impacts on the heart, bone, muscle and other systems; therefore, exercise is always recommended by both the government and society when considering a healthy lifestyle. In aged rats, sedentariness led to a lower SIRT1 level, while treadmill running significantly increased it [141]. The direct effect of exercise might be on the muscles—not only the skeletal muscles but, also, the myocardium. Treadmill running for eight weeks could enhance SIRT1/PGC-1α mRNA expression to improve sarcopenia in aged rats [48]. In addition, exercise also has a role in anti-inflammation. For example, swimming decreased age-related brain apoptosis and inflammatory signaling pathways and increased survival pathway IGF1/PI3K/Akt in the hippocampus [142]. Physical activity augmented SIRT1 pathways to attenuate inflammation in the hearts of d-galactose-induced aged rats by decreasing the FOXO3a level and had benefits on the cell cycle in the heart by decreasing cyclin D and increasing GADD45a mRNA expression in aged rats [143,144,145]. Exercise could also modulate autophagy and lysosome activity. Lysosome function in the brain of wheel running mice were stimulated through AMPK/SIRT1/TFEB by injecting a SIRT1 inhibitor EX527 [146]. Long-term exercise could improve the cardiomyocyte of aged rats histologically and upregulate energy balance-associated proteins such as SIRT1, PGC-1α and AMPK [145]. Furthermore, interval running training has been reported to be a better choice, as it can alleviate age-associated skeletal muscle wasting and bone loss in ovariectomized rats by upregulating the SIRT1 and SIRT6 expression levels to modulate BMPs-induced osteogenesis and osteogenic differentiation [147].

### 3.5. Caloric Restriction and SIRT1 Signaling

Caloric restriction (CR) means a long-term reduction of total caloric intake, usually 60–90% of a normal balanced diet, without being malnourished. CR has positive effects on the body, such as a lower metabolic rate, anti-inflammation and neurovascular protection. Therefore, CR has become one of the recommendations against aging and age-related diseases. Currently, many caloric restriction mimetics (CRMs) have been discovered or invented, including metformin and resveratrol [148,149]. Cohen et al. reported that CR increased the expression of SIRT1, which inhibited stress-induced apoptotic cell death and, thus, improved mammalian cell survival [90]. CR could extend the lifespan by accelerating the elimination of NAD+, leading to the activation of Sir2/SIRT1 [150]. However, other studies showed that the antiaging effect of Sir2 and CR might be parallel, because longer lifespans occur in CR cells without Sir2 [151]. CR alleviated neuropathy and motor deficits through the SIRT1 pathway in MJD mice [46]. In ovariectomized female mice and aged male mice, CR increased the mRNA expression of SIRT1 in bones, which protected them from osteoporosis [152]. Furthermore, a moderate degree of CR at 25% of a normal diet could upregulate the expression of SIRT1 and SIRT3 in the myocardium so that the age-related cardiac degeneration in rats was improved [153].

## 4. MTOR Signaling

The mammalian target of rapamycin (mTOR) is a serine/threonine kinase, which is a part of the phosphoinositide 3-kinase (PI3K)-related kinase family. mTOR functions as an intracellular energy sensor and a central regulator of growth, proliferation, metabolism and aging [154,155,156]. mTOR exists as two distinct protein complexes, mTOR complex 1 (mTORC1) and 2 (mTORC2) [154]. mTORC1 consists of five parts, including mTOR, the regulatory-associated protein of mTOR (Raptor), mammalian lethal with Sec13 protein 8 (mLST8), proline-rich Akt substrate 40 kDa (PRAS40) and DEP-domain-containing mTOR-interacting protein (DEPTOR). mTORC2 contains six elements, including mTOR, rapamycin-insensitive companion of mTOR (Rictor), mammalian stress-activated protein (mSIN1), protein observed with Rictor-1 (Protor-1), mLST8 and DEPTOR. DEPTOR is negatively regulated by both mTORC1 and mTORC2. Inhibiting DEPTOR will activate ribosomal protein S6 kinase (S6K), Akt and SGK1, resulting in cell growth and proliferation (Figure 3) [157].

### 4.1. MTOR Signaling and Aging

The role of mTOR as a central regulator of the lifespan and aging has been studied extensively in the last decade, since mTOR has been linked to many processes that are associated with aging [158,159,160]. Although the underlying mechanisms are still not fully elucidated, the relationship between mTOR signaling and aging has been shown to be conserved from worms to mammals [161,162,163,164]. mTORC1 can be controlled by the tuberous sclerosis complex (TSC)-Rheb pathway and Ras-related GTP-binding protein (Rag) that mediates amino acid signaling [23]. Several studies have indicated that mTORC1 is sensitive to environmental factors such as oxygen, amino acids, growth factors and glucose, regulating many cell processes such as protein translation, autophagy and cell growth [156,165]. On the other hand, mTORC2 is an effector of insulin/insulin-like growth factor-1 (IGF-1) signaling (IIS) downstream of PI3K and is essential for the activation of many kinases. For instance, mTORC2 phosphorylates and activates Akt/PKB, which is a key regulator of cell survival [156,166]. According to the negative effects of mTOR on aging, it was plausible to hypothesize that the expression of mTOR would be increased in the elderly. However, the activity of mTOR and its upstream brain-derived neurotrophic factor (BDNF)/PI3K/Akt signaling was decreased with aging [167]. The disruption of mTORC2 leads to glucose intolerance, diabetes, lower activity level and immunosuppression [156,168]. Therefore, mTORC2 inhibition has negative effects on health and longevity, whereas mTORC1 inhibition extends the lifespan and delays aging. The chronic inhibition of mTOR reduces age-related impairments in spatial learning and memory, but the relationship between mTOR activity and cognitive function follows an inverted U-shaped dose-effect curve [168]. Intestinal flora metabolite trimethylamine-N-oxide (TMAO) inhibits mTOR signaling and, thus, has a negative effect on age-related cognitive dysfunction in SAMP8 and SAMR1 mice through aggravated synaptic damage and the decreased expression of synaptic plasticity-related proteins [169]. A high expression level of DEPTOR was shown to be necessary PI3K and Akt activation in multiple myeloma [157].

### 4.2. Rapamycin and mTOR Signaling

Rapamycin is a natural product secreted by a soil bacterium, *Streptomyces hygroscopicus*, first found as an antifungal antibiotic in 1975 on Easter Island (Rapa Nui), from where it gained its name [170]. Studies have documented that Rapamycin can function as an immunosuppressant, which blocks helper T-cell activation by interfering with signal transduction and can be used for preventing the rejection of organ transplants [171]. In 2009, rapamycin was first shown to extend the lifespan of wild-type mice by the National Institution on Aging Interventions Testing Program [156]. Rapamycin can inhibit mTORC1 by directly blocking substrate recruitment and restricting active-site access through binding the FRB domain of the TOR-associated immunophilin FKBP12 [171]. Rictor and mSIN1 can inhibit the binding of FKBP12-rapamycin to mTOR with Rictor, indicating the rapamycin insensitivity of mTORC2 [172]. Therefore, mTORC1 is acutely sensitive to rapamycin, while mTORC2 is chronically sensitive to rapamycin in vivo and in vitro [166]. Rapamycin was shown to reduce aging markers in human skin [173]. The inhibition of mTOR by dietary rapamycin administration could reverse aging-associated arterial dysfunction [174]. In the early-stage of AD, rapamycin protected the entorhinal cortex and perforate pathway projection from tau-induced neuronal loss, synaptotoxicity, reactive gliosis and neuroinflammation [42]. Houssaini et al. found that rapamycin could rescue cell senescence in chronic obstructive pulmonary disease by inhibiting the mTOR pathway [49]. Moreover, rapamycin restored the senescence phenotype and enhanced the immunoregulatory ability of mesenchymal stem cells of MRL/lpr mice and systemic lupus erythematosus patients [175]. Although rapamycin is an FDA-approved drug, its side effects caused by immunosuppression and the inhibition of mTORC2 may limit its widespread application. For example, the chronic administration of rapamycin induced spermatogenic arrest in adult male mice testis through impairing sex body formation and meiotic sex chromosome inactivation [176]. To overcome these negative effects, other mTOR inhibitors and the adjustment of the dose and administration schedule may be essential. The first generation of mTOR inhibitors like Sirolimus, Temsirolimus and Everolimus, approved by the US FDA, can target mTORC1 through binding to FKBP1. The second generation of mTOR inhibitors, such as NVP-BEZ235, PF-04691502 and OSI-027, act on mTORC1, mTORC2 and PI3K by binding to the kinase domain. The third generation of mTOR inhibitors like RapaLink-1 are bivalent molecules targeting mTORC1 and mTORC2 by binding to the FRB and kinase domains. These third and fourth generations still need to undergo clinical trials to verify their applicability in humans [52,171].

### 4.3. Resveratrol and mTOR Signaling

As described above, resveratrol is a kind of medicine interacting with many molecules involved in aging. In aged mice treated with lipopolysaccharide (LPS), resveratrol inhibited mTOR to increase the expression of proinflammatory genes such as eEIF2α, BIP and ATF4 in response to acute inflammatory stress [177]. Through the Akt/mTOR signaling pathway, resveratrol inhibited glycolysis in human ovarian cancer cells and ameliorated age-related retinal neuropathy in zebrafish [50,178]. Resveratrol induced apoptosis in human U251 glioma cells through the activation of the PI3K/Akt/mTOR signaling pathways [179]. Narasimman et al. reported that resveratrol had a dose-dependent effect on mTOR—namely, a low dose inhibited the phosphorylation of mTOR at serine 2448, while a high dose facilitated the phosphorylation of mTOR at 2481. They also found that resveratrol could upregulate the expression of Rictor in mTORC2 and activate its downstream pathways while weakening mTORC1 activation [180]. Additionally, other phytochemicals have similar effects, such as ginsenoside Rg1, one of the main active ingredients of *Panax ginseng*, which has been found to decrease Akt/mTOR signaling and protect from oxidative stress in a d-galactose-induced subacute aging mouse model [181].

### 4.4. Exercise and mTOR Signaling

Exercise is an activity more complex than what is currently known, as it works on multiple molecules and pathways to result in aging healthily. Aerobic exercise was demonstrated to augment the number of cortical and hippocampal neuronal cells and increase the expression of mTOR in young rat brains [182]. Endurance training regulated the PI3K/Akt/mTOR signaling pathways to prevent renal vascular sclerosis caused by aging [183]. The mechanisms underlying exercise-mediated healthy aging often involve a combination of the AMPK, SIRT1 and mTOR signaling pathways.

### 4.5. Caloric Restriction and mTOR Signaling

CR is a relatively well-known intervention for antiaging up until now, involving several signaling pathways. 4E-BP, a downstream target of mTORC1, was found to be increased and participates in mitochondrial activity and lifespan elongation in the context of CR [184]. mTOR inhibition was involved in BDNF/PI3K/Akt pathway inhibition and further repressed autophagic degradation in the aging hippocampus in the CR group [167]. CR also increased the median telomere length in somatic cells through mTOR signaling [185]. Tulsian et al. reported that CR had a time-dependent effect on mTOR signaling. BMAL1 and CRY are two circadian transcriptional regulators. CR inhibited mTORC1 expression in the liver of BMAL1- or CRY-deficient mice but enhanced mTORC2 capability only in the liver of CRY-deficient mice [186]. CR also protected the skeletal muscle mass in middle-aged rats by regulating mTOR signaling and the ubiquitin-proteasome pathway [187]. Ten months of CR protected against neuronal loss caused by aging and injury and improved learning and memory via the mTOR pathway [188]. Protein restriction (PR) and intermittent fasting (IF), two kinds of CR, inhibited mTORC1 activation and tumor growth in a xenograft mouse model of breast cancer [51]. Moreover, a low protein diet slowed down the nonspecific inflammatory alterations of intestinal and liver cells in aged mice through the mTOR pathway [189]. As for the question about whether CR and TOR inhibition have combined effects, Malene Hansen’s group showed that, under conditions of TOR scarcity, eat-2 *C. elegans*, a type of CR model, did not extend the lifespan, suggesting that their effects were not overlapping [190].

### 4.6. Autophagy and mTOR Signaling

Autophagy is an essential program for maintaining cellular homeostasis [191]. Autophagy induction is a common downstream effect of mTOR, which recycles organelles and is necessary for cells to survive in nutrient deficiency in the context of longevity. The inhibition of mTOR signaling improved autophagy and attenuated the myogenic differentiation in ERCC1 muscle-derived stem/progenitor cells, as well as prevented the cardiac and skeletal muscle function decline in lamin A/C-deficient mice [192,193]. mTORC1 inhibition stimulated transcription factor EB (TFEB) and further activated the coordinated lysosomal expression and regulation (CLEAR) gene network and the transcription of autophagy-related genes (Atgs) [194]. Activation of the mTOR pathway induced VSMCs senescence, which further inhibited autophagy. Autophagy inhibition also enhanced the activation of the mTOR pathway that was induced by adriamycin and increased the number of senescence-associated β-galactosidase-stained cells [195]. Homocysteine, an amino acid metabolite, regulates the connection between mTOR and lysosomal membranes and can inhibit autophagy through activating mTORC1 in vitro and in vivo [196]. PI3K/Akt/mTOR signaling regulated autophagy in macrophages in the development of diabetic encephalopathy and, in particular, chondrocytes by inhibiting miR-20 in osteoarthritis [44,45]. Curcumin, used in Indian and Chinese medicine as a wound-healing agent, among other uses, can also have a beneficial effect on osteoarthritis by possibly enhancing autophagy and reducing cell death and cartilage loss via Akt/mTOR signaling [197]. Trilliun tschonoskii maxin saponin, another natural herb used in Chinese medicine, regulated autophagy by increasing Rheb and decreasing mTOR to improve the learning and memory abilities of d-galactose-induced aged rats [198]. The administration of Geniposide improved cognitive function and reduced amyloid-β plaque deposition in APP/PS1 mice by modulating autophagy via mTOR inhibition [43]. Akt/mTOR signaling inhibition and ERK1/2 activation prevented cancer cell proliferation from glycyrrhizin-induced autophagy and cytotoxicity in HepG2 and MHCC97-H hepatocellular carcinoma cells [199]. Therefore, mTOR has been used as an autophagy inducer in many studies, while autophagy plays a significant role in antiaging.

## 5. Interplay between the AMPK, SIRT1 and mTOR Signaling Pathways

AMPK, SIRT1 and mTOR can also interact with each other, forming an intertwined web. For example, AMPK and SIRT1 increase the expression of Atgs by upregulating FOXOs and PGC-1α and downregulating mTORC1 [194]. Wang et al. showed that the antiaging effects of resveratrol in zebrafish retina involved the activation of the AMPK, SIRT1 and mTOR signaling pathways [178].

### 5.1. Crosstalk between AMPK and SIRT1 Signaling

SIRT1 activates AMPK by stimulating LKB1 via deacetylation, while AMPK activates SIRT1 by enhancing the NAD+ level [200]. Kim et al. found that resveratrol ameliorated oxidative stress and mitochondrial dysfunction by activating the SIRT1/AMPK/PGC-1α axis in an age-related renal injury [201]. Six weeks of exercise, resveratrol and the combination of both significantly improved the expression of p-AMPK and SIRT1 in the brains of aged rats to exert protective effects [202]. Swimming modulated the expression levels of SIRT1/PGC1α, AMPK and FOXO3a in the gastrocnemius muscles of 3-, 12- and 18-month-old rats and inhibited aged hippocampus cell apoptosis and inflammation through IGF1/PI3K/Akt signaling and AMPK/SIRT1/PGC1α signaling [142,143]. Regular aerobic exercise can balance apoptosis and autophagy within the corpus striatum in aged rats through the AMPK/SIRT1 pathway [67]. Wheel running can also activate lysosomal and autophagic functions in the mouse brain via AMPK/SIRT1/TFEB [146]. In SAMP8 mice, a model for aging, wheel running influenced the mitochondrial function via modulating the SIRT1/AMPK pathways [203]. E6155 reduced the fasting glucose and improved the tolerance to oral glucose and insulin through the SIRT1/LKB1/AMPK axis [47]. A moderate dose of resveratrol improved the mitochondrial biogenesis and function via AMPK stimulation induced by SIRT1 [200]. The AMPK/SIRT1/FOXO1 axis also participated in modulating apoptosis in bovine intracellular adipocytes [204]. Glucose restriction regulated nicotinamide phosphoribosyltransferase (NAMPT), an important rate-limiting enzyme in the synthesis of NMN, to stimulate SIRT1 with the help of AMPK to have a negative effect on skeletal myoblast differentiation [205].

### 5.2. Interaction between AMPK and mTOR Signaling

AMPK interacts with mTOR in two main ways, by directly phosphorylating Raptor and indirectly phosphorylating TSC2, resulting in the activation of the GTPase-activating protein and the combination of Rheb on lysosomal membranes [206,207]. A moderate dose of bilberry anthocyanin (MBA) upregulated the expression of OCLN, ZO-1 and autophagy associated-proteins ATP6 V0C, ATG4D and CTSB to induce autophagy through the AMPK/mTOR pathway and then improved the intestinal epithelial barrier function and oxidative stress resistance effects in aging female rats [208]. Genistein dose- and time-dependently activated LKB1/AMPK/mTOR signaling to induce autophagy, which protected VSMCs from aging [209]. Acarbose competitively and reversibly inhibited salivary and pancreatic α-amylases and small intestine brush border α-glucosidases, which then decreased postprandial glucose and subsequently modulated the AMPK- and mTORC1-regulated metabolism pathways, leading to the improved survival of Apc mice, a mimic of familial adenomatous polyposis in humans [210]. Metformin improves cartilage degeneration in osteoarthritic mice through the AMPK/mTOR pathway [211]. Qing et al. found that metformin regulated the AMPK/mTOR/NACHT, LRR and PYD domains-containing protein 3 (NLRP3) inflammasome signaling pathways to induce M2 macrophage polarization, leading to the promotion of wound healing [212]. In osteoarthritic mice, a local intra-articular injection of resveratrol induced autophagy through AMPK/mTOR signaling and attenuated cartilage degeneration [213]. Exogenous hydrogen sulfide (H2S) rescued cardioprotection from ischemic postconditioning by stimulating autophagy via the AMPK/mTOR pathway in isolated aged rat hearts and aged cardiomyocytes [214]. Oleuropein aglycone (OLE), the main polyphenol in extra virgin olive oil, exerted beneficial health effects by activating autophagy through the AMPK/mTOR pathways in cultured neuroblastoma cells and OLE-fed mice [215]. CRMs ameliorated hyperglycemia-induced senescence and epithelial–mesenchymal transition (EMT) by activating AMPK/mTOR signaling [216]. In ApoE-deficient mice, CR upregulated Fgf21 to stimulate AMPK/mTOR signaling, leading to a decrease in the formation of neurofibrillary tangles by inhibiting tau phosphorylation [217].

### 5.3. Interplay between SIRT1 and mTOR Signaling

The relationship between SIRT1 and mTOR remains largely unexplored. It was found that SIRT1 inhibits mTOR signaling by interacting with TSC2 [218]. SIRT1 was needed for rapamycin to have an effect on high-glucose-induced mesangial cells senescence [219]. The mTORC1 inhibitor rapamycin could enhance the expression levels of SIRT1 and AMPK and improve the deacetylase ability of SIRT1 in AML12 hepatocytes [220]. mTOR and SIRT1 cooperated to promote the proliferation of intestinal stem cells in CR [221]. SIRT1 decreased the autophagy and mitochondrial function in embryonic stem cells by downregulating the mTOR pathways in response to oxidative stress [222]. Similarly, in alcoholic liver disease mice and patients, the lack of DEPTOR and SIRT1 induced the abnormal activation of mTORC1, resulting in an increased phosphorylation level of mTOR and S6K1 to aggravate inflammation and acute-on-chronic liver injuries [223].

## 6. Summary

Aging may be an irreversible process with complex mechanisms. Therefore, the real target of antiaging may not be to rejuvenate but to accomplish healthy aging, which means ameliorating the physiological decline and dysfunction in late life. Heathy aging may be attained when we have a better understanding of the mechanisms underlying aging. Since Harmen first interpreted the free radical theory for aging in 1956, many aging theories have been proposed, such as the mitochondrial theory, where mitochondria-related oxidative damage can be involved in the aging process. Thus, up to now, the known hallmarks of aging included genomic instability, the loss of proteostasis, epigenetic alteration, telomere attrition, deregulated nutrient sensing, mitochondrial dysfunction, cell senescence and stem cell exhaustion [224,225]. Recently, we proposed a microcirculatory theory of aging [226]. Indeed, aging is regulated by specific signaling pathways, and among them, the AMPK, SIRT1 and mTOR pathways play critical roles. They can function individually or in combination to affect the aging process. More specifically, metformin, resveratrol and exercise can activate AMPK, while AMPK can inhibit autophagy. Resveratrol, NAD+, exercise and CR increase the SIRT1 expression. Rapamycin, resveratrol, exercise and CR perturb the mTOR and impact autophagy. Additionally, AMPK can be promoted by SIRT1 and mTOR, SIRT1 can be activated by AMPK and mTOR and mTOR can be inhibited by AMPK and SIRT1 (Figure 4). The identification of these aging signaling pathways opens a new avenue to develop promising targets for heathy aging by stimulating the longevity pathways or inhibiting the aging pathways.

## Figures and Tables

**Figure 1 cells-10-00660-f001:**
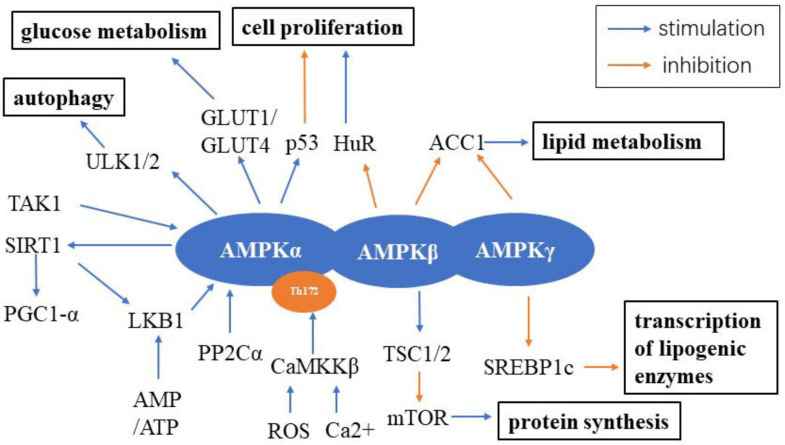
AMPK signaling in the cell. AMPK is composed of three subunits: the catalytic α subunit and regulatory β and γ subunits. The upstream of AMPK includes LKB1, CaMKKβ, TAK1 and PP2Cα, while the downstream pathways controlled by AMPK include ULK1/2, SIRT1, TSC1/2, SREBP1c, ACC1, HuR, p53 and GLUT1/GLUT4. Abbreviations: AMPK: adenosine onophosphate-activated kinase; LKB1: liver kinase B1; CaMKKβ: Ca^2+^/calmodulin-dependent protein kinase kinase β; TAK1: transforming growth factor β-activated kinase 1; PP2Cα: protein phosphatase 2Cα; ACC1: aminocyclopropane-1-carboxylic acid; GLUT1: glucose transporter 1; GLUT4: glucose transporter 4; ULK1/2: autophagy activating kinase 1/2; PGC1-α: peroxisome proliferator-activated receptor gamma coactivator-1α; ROS: reactive oxygen species; SIRT1: Sirtuin 1; AMP: adenosine monophosphate; ATP: adenosine triphosphate; TSC1/2: tuberous sclerosis complex 1/2 protein; mTOR: the mammalian target of rapamycin; SREBP1c: sterol regulatory element binding protein 1c; HuR: human antigen R.

**Figure 2 cells-10-00660-f002:**
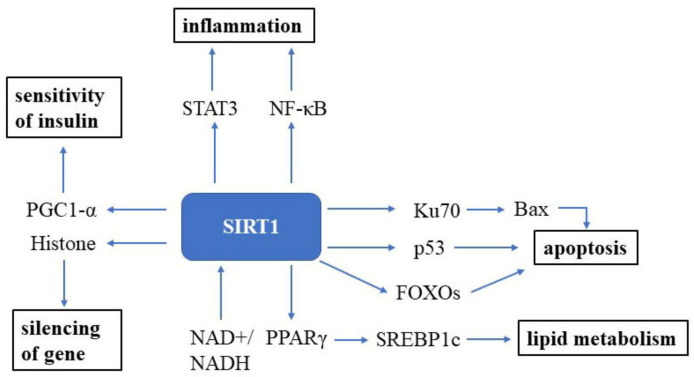
SIRT1 signaling in the cell. SIRT1 is a well-known NAD+-dependent deacetylase that impacts several molecules to promote health. Abbreviations: STAT3: signal transducer and activator of transcription 3; NF-κB: NF-kappa B; Bax: bcl-2-associated X protein; FOXOs: forkhead box transcription factors; PPARγ: peroxisome proliferator-activated receptors; NAD+: oxidized nicotine adenine dinucleotide; NADH: reduced nicotine adenine dinucleotide.

**Figure 3 cells-10-00660-f003:**
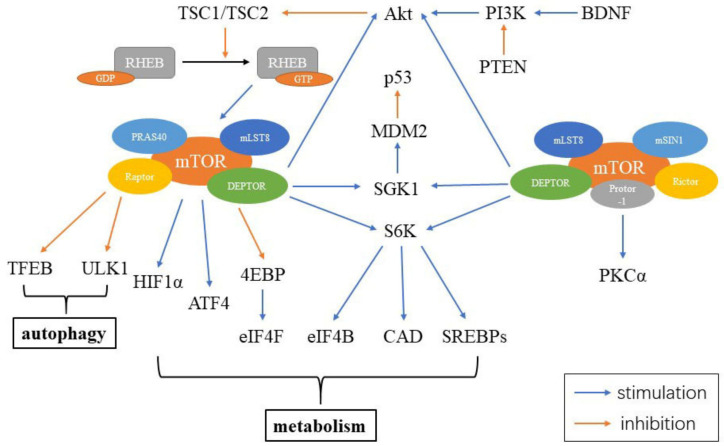
Constituents of mTORC and mTOR signaling in the cell. mTORC1 includes mTOR, Raptor, mLST8, PRAS40 and DEPTOR, while mTORC2 contains mTOR, Rictor, mSIN1, Protor-1, mLST8 and DEPTOR. Abbreviations: RHEB: ras homolog enriched in brain; GDP: guanosine diphosphate; GTP: guanosine triphosphate; PRAS40: proline-rich Akt substrate 40 kDa; mLST8: mammalian lethal with Sec13 protein 8; Raptor: the regulatory-associated protein of mTOR; DEPTOR: DEP-domain-containing mTOR-interacting protein; TFEB: transcription factor EB; HIF1α: hypoxia inducible factor 1α; ATF4: activating transcription factors 4; 4EBP: 4E binding protein; eIF4F: eukaryotic initiation factor 4F; eIF4B: eukaryotic initiation factor 4B; CAD: carbamoyl-phosphate synthetase; S6K: S6 kinase; SGK1: glucocorticoid induced protein kinase 1; MDM2: murine double minute 2; Akt: protein kinase B; PI3K: phosphoinositide 3-kinase; PTEN: phosphatase and tensin homolog; BDNF: brain-derived neurotrophic factor; mSIN1: mammalian stress-activated protein; Protor-1: protein observed with Rictor-1; Rictor: rapamycin-insensitive companion of mTOR; PKCα: protein kinase Cα.

**Figure 4 cells-10-00660-f004:**
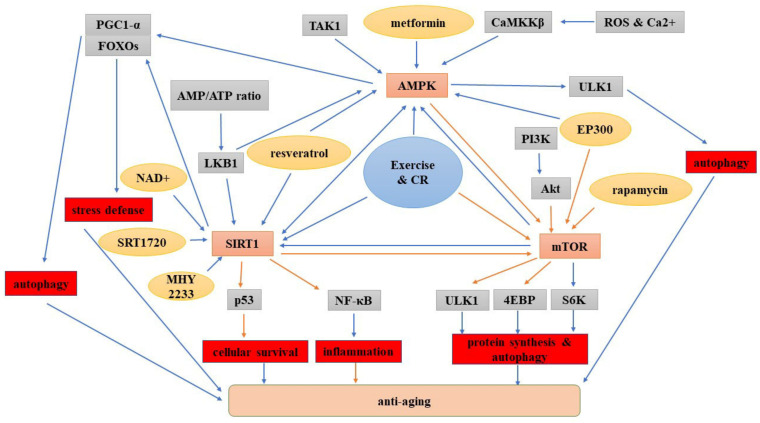
Illustration of the roles of the AMPK, SIRT1 and mTOR signaling pathways in aging. The blue arrows indicate a positive effect, and the orange arrows indicate a negative effect. Abbreviations: CR: caloric restriction; NF-κB: NF-kappa B.

**Table 1 cells-10-00660-t001:** Age-related diseases and their currently identified signaling pathways.

Disease	Signaling Pathways	Cell Type/Model	Reference
Hutchinson-Gilford progeria syndrome (HGPS)	AMPK	HGPS cells	[30]
Muscle atrophy	protein kinase A (PKA)/LKB1/AMPK	High-fat diet-induced muscle atrophy in aged animal model	[39]
Parkinson’s disease (PD)	AMPK, brain-derived neurotrophic factor (BDNF)	6-OHDA-induced PD animal model; MPTP-treated and haloperidol-induced catalepsy animal models	[37,38]
Alzheimer’s disease (AD)	SIRT1, mTOR	5xFAD-ApoE4 (E4FAD) AD mouse model; Htau mice; adeno-associated viral vector-based mouse model of early-stage AD-type tauopathy; APP/PS1 mice	[40,41,42,43]
Diabetic encephalopathy (DE)	phosphoinositide 3-kinase (PI3K)/protein kinase B (Akt)/mTOR	Streptozotocin (STZ)-induced rat model	[44]
Osteoarthritis (OA)	PI3K/Akt/mTOR	Human OA chondrocytes	[45]
Machado-Joseph disease (MJD)	SIRT1	Machado-Joseph disease mouse model	[46]
Diabetes	SIRT1/LKB1/AMPK	Type-2 diabetic KKA mice	[47]
Sarcopenia	SIRT1/PGC1-α	Aged rats	[48]
Chronic obstructive pulmonary disease (COPD)	mTOR	Lung tissue and derived cultured cells from patients with COPD	[49]
Ovarian cancer	Akt/mTOR	Human ovarian cancer cells	[50]
Breast cancer	mTOR	Xenograft mouse model	[51]

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
