# Peer review of "Key Signaling Pathways in Aging and Potential Interventions for Healthy Aging"

_cells, 2021, doi:10.3390/cells10030660_

Round 1
Reviewer 1 Report
This is a comprehensive review and nicely written paper to summarize the current knowledge and progress in the area of aging and related disorders. The review covers broad topics and provides useful information to readers including research scientists, physician, educators and general public. The only suggestion I have is to add a paragraph about the relationship of sex hormone and their influence on aging in male and female. This is an important research area that may be interesting to many readers.
Author Response
Comment 1: This is a comprehensive review and nicely written paper to summarize the current knowledge and progress in the area of aging and related disorders. The review covers broad topics and provides useful information to readers including research scientists, physician, educators and general public. The only suggestion I have is to add a paragraph about the relationship of sex hormone and their influence on aging in male and female. This is an important research area that may be interesting to many readers.
Response: Thank you very much for your time involved in reviewing the manuscript and your very encouraging comments on the merits. The paragraphs about the relationship of sex hormone and their influence on aging in male and female has been added on page 3, lines 107-115 and lines 19-122; page 8, lines 314-323.
Reviewer 2 Report
Manuscript number: cells-1121458
Title: Key signaling pathways in aging and potential interventions for healthy aging
Comments to the Author
In this manuscript, authors reviewed recent breakthroughs in the mechanistic understanding of biological aging, focusing on AMPK, SIRT1 and mTOR pathways, which are currently considered critical for aging. They also described how these proteins and pathways may potentially interact each other to regulate and how knowledge of these pathways may lead to new interventions for anti-aging and against age-related disease.
In general, the manuscript is well organized. I have some comments that I believe might help the authors in increasing the impact of this manuscript.
Comments
- Authors indicated the effect of resveratrol on these signaling pathways. Recommend to introduce other phytochemicals, which show similar effect on aging through regulation of these signaling pathways.
- If there are clinical studies for anti-aging using (phyto)chemicals or medicines, addressed the studies in separate section.
- Recommend to add table which including all results according to disease, signaling, cell type, or model.
- There are some typos. Correct all.
Author Response
Comment 1: Authors indicated the effect of resveratrol on these signaling pathways. Recommend introducing other phytochemicals, which show similar effect on aging through regulation of these signaling pathways.
Response: Thanks for your great suggestion on improving the accessibility of our manuscript. We have added other phytochemicals in ‘4.3. resveratrol and mTOR signaling’ and ‘5.2. interaction between AMPK and mTOR signaling’ such as ginsenoside Rg1 and acarbose. Please see page 13, lines 555-559; page 16, lines 666-671.
Comment 2: If there are clinical studies for anti-aging using (phyto)chemicals or medicines, addressed the studies in separate section.
Response: Thanks for your advice on improving the accessibility of our manuscript. The clinical studies have been added in ‘2.2. Metformin and AMPK signaling’ on page 4, line 151-154.
Comment 3: Recommend adding table which including all results according to disease, signaling, cell type, or model.
Response: The reviewer’s comments are well taken. Table 1 has been added accordingly, which is located on page 5.
Comment 4: There are some typos. Correct all.
Response: We have carefully and thoroughly proofread the manuscript. The typos have been corrected.
Reviewer 3 Report
In the manuscript authors described key signalling pathways in ageing and potential interventions for healthy ageing. The manuscript is written fluently and precisely, but I suggest some corrections before the publication. The comments are written below.
-The manuscript needs English editing before the publication.
-The authors should also mention other drugs with potential anti-ageing effects, such as AGE cross-links blockers, etc.
-Figure 1 should be corrected and upgraded to become more explanatory and narrative-the process of ageing should be included, other potential anti-ageing agents could be also mentioned (also agents that are not described in the manuscript), etc.
Author Response
Comment 1: The manuscript needs English editing before the publication.
Response: The paper has been edited by a native English speaker.
Comment 2: The authors should also mention other drugs with potential anti-ageing effects, such as AGE cross-links blockers, etc.
Response: Thank you for your advice. We have added other phytochemicals like
Pyridoxamine, 17β-estradiol, ginsenoside Rg1 and acarbose. The relevant contents can be found on page 8, lines 314-324; page 13, lines 555-559 and page 15, lines 666-671.
Comment 3: Figure 1 should be corrected and upgraded to become more explanatory and narrative-the process of ageing should be included, other potential anti-ageing agents could be also mentioned (also agents that are not described in the manuscript), etc.
Response: According to the reviewer suggestion, we have modified the Figure 1.